# A Machine Learning-Based Intrusion Detection System for IoT Electric Vehicle Charging Stations (EVCSs)

Mohamed ElKashlan [1], Mahmoud Said Elsayed [2,*] , Anca Delia Jurcut [2] and Marianne Azer [3]

1 School of Information Technology and Computer Science, Nile University, Cairo 12677, Egypt
2 School of Computer Science, University College Dublin, D04 V1W8 Dublin, Ireland
3 National Telecommunication Institute, Nile University, Cairo 12677, Egypt
* Correspondence: mahmoud.abdallah@ucdconnect.ie

**Abstract:** The demand for electric vehicles (EVs) is growing rapidly. This requires an ecosystem that meets the user's needs while preserving security. The rich data obtained from electric vehicle stations are powered by the Internet of Things (IoT) ecosystem. This is achieved through us of electric vehicle charging station management systems (EVCSMSs). However, the risks associated with cyber-attacks on IoT systems are also increasing at the same pace. To help in finding malicious traffic, intrusion detection systems (IDSs) play a vital role in traditional IT systems. This paper proposes a classifier algorithm for detecting malicious traffic in the IoT environment using machine learning. The proposed system uses a real IoT dataset derived from real IoT traffic. Multiple classifying algorithms are evaluated. Results were obtained on both binary and multiclass traffic models. Using the proposed algorithm in the IoT-based IDS engine that serves electric vehicle charging stations will bring stability and eliminate a substantial number of cyberattacks that may disturb day-to-day life activities.

**Keywords:** anomaly detection; cyber security; feature selection; Internet of Things (IoT); intrusion detection system (IDS); machine learning; security



## 1. Introduction

The IoT paradigm has recently become part of the daily activities of our lives. However, the vulnerabilities and insecurity associated with introducing IoT devices has alerted IoT network operators and users [1]. Electric vehicle charging station deployment, as a part of smart cities, has become popular in the last few years. Many countries have ambitious plans to adopt many EVCSs quickly [2]. These new charging stations benefit from IoT technology enhancements and provide intelligent features that make life easier and provide more controls to EVCS operators. The EVCS, as an IoT device, cannot be decoupled from the Internet. This is to offer comprehensive services to the customers. Unfortunately, this enables a set of cyber-attacks against the whole EVCS ecosystem. The effect is not limited to EVCSs alone. It extends to the critical infrastructure of the power grid and end users equally. The EVCS, the power grid, and the end users are the main components of the EVCS ecosystem. All these components of the EVCS ecosystem are subject to different kinds of IoT cyber-attacks [3]. The EVCS industry requires rapid development in the infrastructure to support sustainable growth. This mandates the construction of a reliable charging station ecosystem for electric vehicles. The IoT ecosystem powers the rich data from electric vehicle stations. This allows the developers to offer more capabilities that benefit the end user, such as remote monitoring and user accounting. One of the valuable capabilities that can benefit the user is the remote scheduling of EV charging based on the reduced electricity tariff at night. One of the significant problems in IoT cybersecurity is that the malicious traffic passing through the IoT system cannot be easily identified and separated from legitimate traffic. This is because advanced attackers use sophisticated methods to evade

detection. Contrary to IT security, IoT security is more critical than traditional IT network security. This is due to its massive attack surface and multiplied weak-spots due to the amount of data shared between the nodes and constant connectivity to serve users [4]. The main players in detecting malicious traffic, in both IT and IoT systems, are intrusion detection systems, which undergo continuous enhancements to improve their efficiency and accuracy. The key to success in evaluating any IDS system is using a relevant dataset [5]. Intrusion detection is a hot topic in academia; the usage of machine learning (ML) and deep learning (DL) algorithms have given traction to the development of more efficient and precise detection methods for different types of cyberattacks when using IDSs [6]. In a recent study [7], the cost of a data breach in the energy sector was shown to have increased dramatically in the past few years.

To address these challenges, the use of machine learning (ML) techniques has become increasingly important in detecting and preventing cyberattacks on EVCS. ML algorithms are well suited for this task due to their ability to learn from large amounts of data and make predictions based on that learning. With the increasing use of EVs and EVCSs, there is a growing need for effective and efficient methods to detect and prevent cyberattacks.

In this paper, we propose the use of ML algorithms for detecting cyberattacks on EVCSs. The methodology involves collecting and preprocessing data from various sources including the charging station's network traffic, the charging process, and the charging station's environment. These data are then used to train and evaluate various ML models for detecting and classifying potential cyberattacks. Our study compares the performance of different ML algorithms and provides insights into the most effective methods for detecting cyberattacks on EVCSs.

In this paper, we define the EV charging ecosystem, starting from the EVCS, passing through the communication and transportation protocol, and ending with the management system, the EVCSMS. We describe the major attacks in each component, the main attack vectors, and the different vulnerabilities affecting the ecosystem. This includes attacks on the charging stations, the user, and the most destructive type of attack that can occur, which is on the power grid.

Moreover, from the detection side, we explain the use of machine learning methods in anomaly detection and how we use the IoT dataset to represent the traffic. The used IoT dataset represents typical traffic in the IoT systems, considering the EVCS as a viable example of an IoT system. The dataset represents the traffic and attacks the EVCS can suffer from. That is why we use a native IoT dataset, which is the IoT-23. It is derived from real IoT devices. In addition, for the sake of the algorithm agility, the irrelevant redundant features found in the dataset are first eliminated. This increases the accuracy and solves the overfitting and underfitting problems commonly seen in these models. By exploring the use of ML in detecting cyberattacks on EVCS, we hope to contribute to the development of more secure and reliable charging infrastructure for EVs.

The contributions of this paper are as follows:

(1) Apply different machine learning classifying algorithms to identify the malicious traffic in EVCSs using a native IoT dataset.
(2) Represent major attacks and vulnerabilities in EVCSs and the efforts carried out in the literature to mitigate them.
(3) Use machine learning algorithms, that were originally used in tackling different non-IoT security problems, on an IoT security problem.
(4) Identify malicious traffic using a limited amount of trained data through a reduced dataset.

The remainder of the paper is organized as follows: Section 2 describes the context of the problem. In Section 3, we demonstrate the related work carried out in the literature for securing IoT systems using ML- and DL-based IDSs. Section 4 presents the methodology and components used in the simulation to mimic the real problem. The experimental results in Section 5 demonstrate the results of the different machine learning classifier algorithms. Finally, Sections 6 and 7 discuss the limitations and conclude the paper, respectively.

## 2. Electric Vehicle Charging Station Background

The EVCS system is still fresh and has a wide surface of attacks. Therefore, it is the ideal target for exploitation by state-sponsored actors and competitors. With the increasing number of EVCS units connected to the Internet, the adversary can take advantage of the vulnerabilities in the system and compromise public and private EVCSs. Exploitation can be executed remotely using the Internet or locally through the local area network (LAN) if there are weak access controls in the EVCS network.

The EVCS involves three main components: sensing, networking, and communication layers. The most vulnerable part, which attackers are interested in, is communication and networking. This layer handles communication with the supervisory control and data acquisition system (SCADA). In addition, this layer ensures proper communication between the EVCS and the end user via the Internet. This can be achieved using various technologies such as Bluetooth, Wi-Fi, cellular, and even wired digital subscriber lines (DSL) or fiber optics. The other components of the EVCS, such as the sensing and computational components, are considered internal systems. They are also vulnerable but need local interaction with the EVCS or can be used in further steps after compromising the EVCS via the communication and networking component.

Additionally, the attacker can form a silent botnet from a number of compromised EVCSs to conduct distributed attacks against other systems. A study that estimates the cyber insurance cost against cyberattacks on EVCSs [8] mentioned that the financial losses due to cyberattacks on EVCS systems are hefty. Figure 1 depicts the different components in the EVCS ecosystem.

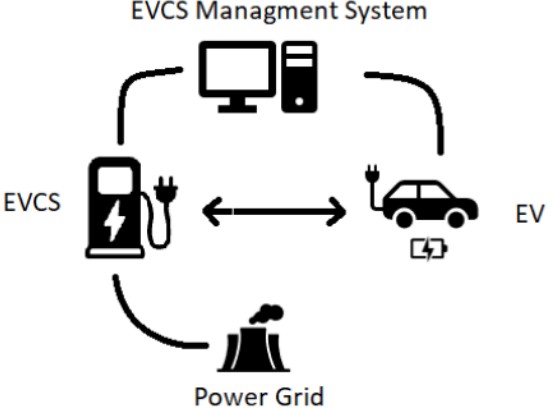

**Figure 1.** EVCS ecosystem.

The attacks on the EV charging system can be classified into several categories, such as attacks affecting the EVCS itself, attacks targeting the end user, or attacks on the power grid that provides electricity to the entire charging stations.

(1)  **Attacks against the EVCS:** One of the main vulnerabilities in the EVCSMS is cross-site scripting (XSS), where the input sanitization is not maintained properly. This enables the attacker to compromise the EVCS, obtain its charging schedule, and control the entire charging process. Similarly, the SQL injection (SQLi) is of the same severity and can enable the attacker to access the EVCSMS database, which includes the users' records and the administrator credentials. This allows the attacker to further perform illegal activities by modifying the charging costs. Depending on the scale of the compromised EVCS, an attacker can form a botnet to launch other denial of service attacks (DoS). It is also possible to deny the end user that uses the EVCS from using the EVCS by continuously restarting the station, which disrupts the charging operation.

(2)  **Attacks against the users:** The most destructive effect on the users is to damage the EV battery. Despite the safety systems in electric vehicle onboard chargers, the attacker can severely damage the battery by increasing/decreasing the voltage and

current significantly above the safety limits. Moreover, depending on the adversary's motivation towards the attack, organized espionage can occur, such as obtaining a user's personal identifiable information (PII). This can include the full details of their username, address, and contacts, which can be further used in other attacks on a specific user. As the EVCS manages the financial accounts for the users, a man-in-the-middle attack can store the user's financial information. Lately, attackers use this financial information for fraud or sell the data of users on the dark web, which is the marketplace for this type of stolen information.

(3) **Attacks targeting the power grid:** As the power grid's operation involves serving large scale operations of millions of subscribers, it is the most critical infrastructure component directly affecting the daily life of many users and the industry. This can cause destructive damage and has significant implications for the economy [9]. The electric supply-and-demand balance is crucial to the stability of the power grid system; by tampering with this balance, the system's stability is torn down. An attacker who gains control of the EVCSMS can launch a synchronized charging and discharging of the compromised EVCS. The charging process flows from the power grid to the EVCS, and the discharge is a feature in the EVs that is also called vehicle-to-grid (V2G). The attacker can perform charging and discharging quickly to cause a disturbance by sudden switching. This leads to frequency overshooting, in which the generated power is more than the load power. Accordingly, the system tries to regain its stability by reducing the generated power, and hence the frequency increases. The attacker responds by increasing the load in a short time. This causes imbalance and reduction in the frequency and so on. The safety systems of the power plants step in at this stage and isolate the entire power plant to avoid further damage to the generators. Reconnecting a power plant to the grid may take days, which is considered a severe interruption to the economy.

## 3. Related Work

Researchers in both academia and the industry have focused, in the past few years, on applying artificial intelligence (AI) in the IDSs of IoT systems. This combination has proven to be effective in the anomaly detection of malicious activities. For instance, machine learning algorithms such as naïve Bayes (NB), logistic regression (LR), and decision tree (DT) have been widely used in the detection of different types of network-based attacks. The mentioned algorithms basically rely on learning from predefined features, which is a typical characteristic of IoT traffic. Other research has focused on deep learning methods to achieve better accuracy and bypass feature selection. The algorithm and the dataset used in the model are both of a key role in obtaining accurate results that can be applied in real-world scenarios.

A study that was made [10] examined sixteen different EVCSMSs deployed by reputable vendors in the industry, which was deployed mainly in Europe and the United States and found critical zero-day vulnerabilities in the web, mobile, and firmware. Moreover, they used the discovered vulnerabilities to compromise the EVCS, causing instability in the power grid as a practical implication against the EVCS and the users. EVCS firmware and EVCSMS security has received little attention from academia compared to security research on the other components of the EV ecosystem. This was the first study on the vulnerabilities in practical EVCS, which shows the severe impact that can be felt if successfully exploited. The results show that the live systems used currently in charging the EVs are highly vulnerable and can be attacked easily, causing wide damage to the power grid and the users. The study proposed at the end a set of mitigations that can be applied to the vulnerable systems to decrease the effect of these attacks.

In a previous study [11], authors proposed an intrusion detection system using the deep belief network (DBN). DBN is an algorithm for augmenting different unsupervised networks stacked together to serve as an input for the next layer. This is carried out by using autoencoders, specifically restricted Boltzmann machines (RBMs). After the training,

the target classification, as per their model, is to have an output of 0, i.e., no intrusion detected, or 1, which means intrusion detected. The used dataset in this model is the TON_IOT dataset, which contains 30,000 tuples. This IoT dataset was obtained from a practical depiction of a medium-scale network at the UNSW Canberra Cyber Range and IoT labs (Australia). This dataset contains various IoT attacks in a nonuniform way, which is considered a reliable source. They implemented the model using TensorFlow. The results showed that the accuracy of this model reached 86% and an F1 score of 84%. The interesting part of this study is that it compares the results of the DBN with other algorithms, which shows a low accuracy for the DBN (86%) compared to the DNN (96% accuracy) and LSTM + CNN (97% accuracy) algorithms. In comparison, the DBN performed better than NB (54% accuracy) and SVM (60% accuracy).

A comprehensive survey on the machine learning and deep learning methods used in the intrusion detection systems for the IoT was made by Thakkar et al. [12]. They listed the security issues and challenges facing IoT systems. To tackle the risks of the open communication layer in the IoT system, the authors in [13] proposed a deep learning-based IDS to detect DoS attacks within the EVCS. Deep neural network (DNN) and long short-term memory (LSTM) algorithms were implemented. While both methods achieved 99% accuracy, the LSTM was superior according to their study in terms of accuracy, precision, and recall. While the results were promising, they focused on the DDoS attack. In our opinion, using the deep learning method in a real-time problem is a resource-intensive method that is not practical and may detect malicious activity after passing the traffic to the network. Lastly, the dataset used (CICIDS 2018) is not derived from IoT-native data, which makes it irrelevant for application to the IoT problem.

In another study [14], authors used the IoT-23 dataset effectively and applied different machine learning algorithms. Random forest (RF), naïve Bayes (NB), multi-layer perception (MLP), support vector machine (SVM), and AdaBoost (ADA) were evaluated. The random forest algorithm achieved the best accuracy of 99.5% among the others. While the methodology and the data samples were different from our study, as we used highly randomized data, the results for the above study, along with the other study of Thamataiselvi et al. [15], was in line with our results too.

The authors in [16] studied the different types of attacks targeting IoT systems, including authentication, access control, secure offloading, and malware detection. In addition, they made a detailed study on the different types of machine learning algorithms, including supervised, unsupervised, and reinforcement learning (RL), but they focused on implementing these algorithms on the limited resources of IoT devices. They used the KDD CUP 99 dataset, which we see was not suitable for representing modern IoT devices.

The survey conducted in [17] focused on IoT–ML in healthcare and focused mainly on the sensing layer, which they see as the most vulnerable layer in the IoT ecosystem. However, the authors did not touch the issue of the dataset used by the ML algorithms in the survey.

A benchmark study on anomaly detection in ML based IDS [18] evaluated different supervised and unsupervised ML algorithms (ANN, DT, k-NN, NB, RF, SVM, CNN, EM, K-means, and SOM) using the CICIDS 2017 dataset; the models show limitations in detecting novel types of attacks with multiclassification.

A recent review on the IDS based on ML and DL algorithms [19] used different datasets. The combination of the ML and DL with the different datasets generated promising results to achieve the highest accuracy. However, the F1 and recall was not the focus of the study, which is not a fair evaluation, in our opinion. The comprehensive study on the IoT IDS conducted in [20] showed clearly that the computational power of the DL is impractical for use in a restrained environment such as the IoT. However, pushing this capability to the edge or cloud can partially solve this problem. Similar to our study, the classifiers play an essential role in achieving two important goals, which are performance and accuracy. In Table 1, we compare the state-of-the-art research in the same field during the past years.

**Table 1.** Comparison between the ML–IoT studies carried out in the literature.

| Ref | Description | Advantages | Disadvantages | Dataset Used | Conclusion |
|---|---|---|---|---|---|
| [10] | Comprehensive study on 16 vendors of EVCSs, showing the vulnerabilities on each vendor | Examined 16 EVCSMS vendors | No proposed solution provided to detect these attacks and focused only on the remediation of the vulnerabilities | - | Most of the EVCSMS are highly vulnerable |
| [11] | Deep learning algorithms evaluation using TON_IOT dataset | Deep learning was used | Focusing on DL only | TON_IOT | High accuracy for the DNN algo—96% |
| [12] | ML and DL algorithm evaluation to mitigate IoT attacks | Survey on both ML and DL methods in IoT-IDS | No specific dataset | - | Detection methods for IoT do not address wide range of attacks |
| [13] | Deep learning method for detecting DDoS attacks using the CICIDS 2018 dataset | Deep learning-based IDS was proposed | Focused only on DDoS attacks | CICIDS 2018 | LSTM was superior, according to their study in terms of accuracy, precision, and recall |
| [14,15] | ML-based IoT IDS using IoT-23 dataset | Examined different ML methods | Data not randomized | IoT-23 | Random forest (RF) algorithm achieved 99.5% accuracy |
| [16] | Benchmarking different supervised, unsupervised, and reinforcement learning (RL) algorithms on different types of attacks using KDD CUP 99 dataset | Studied different types of attacks versus different detection ML algorithms | Dataset is not suitable for the application | KDD CUP 99 | The intrusion detection schemes based on unsupervised learning algorithms sometimes have misdetection rates that are nonnegligible for IoT systems |
| [17] | A survey on the IoT ML Based Healthcare system | Studied IoT ML in healthcare | Focus on the sensing layer | - | Identified a number of research challenges, including exploration of deep learning-based models |
| [18] | Benchmarking of machine learning for anomaly-based intrusion detection systems in the CICIDS2017 dataset | Studied different supervised and unsupervised ML algorithms | Focus on web attacks only | CICIDS 2017 | Experiment results show the absence of any single ML algorithm that are able to detect all types of web attacks |
| [19] | A review on intrusion detection systems based on machine learning algorithms | Different ML and DL algorithms | Focus mainly on the accuracy | Different datasets | The application and type of dataset have a great influence on the accuracy |
| [20] | A comprehensive deep learning benchmark for IoT IDS | Comprehensive study on different ML and DL algorithms | DL algorithms needs substantial amount of computation power, which is impractical to IoT | Different datasets | Selecting the classifier is important to get a good performance and accuracy |

## 4. Experimental Methodology

In this section, we discuss the methods used in building the model by examining different classifiers and the chosen dataset representing a typical IoT traffic.

The data are divided into two samples: a training sample that represents 90% of the data, and testing sample, which represents 10% of the data. The training sample undergoes pattern analysis and then label-learning using the machine learning process. While in the testing phase, the labels are hidden from the model, and the classification engine classifies the unlabeled data based on the machine learning process that was carried out in the training phase. The outcome of the classification process in the testing phase is the labels being predicted by the learned model.

Next, we describe the dataset used in the problem simulation and how we selected some features to have reduced lightweight training data to be used in the training model without affecting the accuracy.

### 4.1. Dataset Description

The IDS validation depends mainly on the datasets used in the evaluation. Simulating intrusive behavior allows us to evaluate the IDS's capability. However, due to privacy reasons, it is not easy to obtain real traffic for commercial products. The available datasets that were developed, by time, include KDD, DRPA, NDS–KDD and ADFA–LD. They are used by different researchers for benchmarking. We used the most recent IoT dataset, which is IoT-23.

IoT-23 [15] is the newest dataset, derived from network traffic generated by real commercial IoT devices. The dataset consists of twenty malware traffic captures collected from the IoT devices and three traffic captures for normal (benign) traffic. The dataset was first published at the beginning of 2020 by Stratosphere Lab in the Czech Republic, funded by Avast Software, Prague.

### 4.2. IoT-23 Dataset

The use of machine learning and deep learning in anomaly detection has been common in research papers [21] over the past decade. These efforts sometimes lack the correct adequacy of the chosen dataset to the problem. The IoT-23 dataset is composed of twenty scenarios, which represent the malicious traffic of different types of attacks on the IoT network. Each scenario has the malware name that was executed on the IoT device. Moreover, it is composed of three scenarios with normal IoT traffic (not infected), which are used as a reference to the normal traffic. The number of total scenarios is twenty-three, which is why it is named IoT-23. The 23 scenarios were running in a controlled environment and connected to the Internet, like any typical IoT device. Hence, the IoT dataset contains two types of traffic, the normal and malicious traffic scenarios available to the community. This secures the evaluation of the binary classification. However, the malicious traffic flows are further labeled with extra labels to be able to have a multiclass classification. Below is a brief explanation on the multiclass labels found in the malicious flows.

(1) **Benign:** The benign tag shows that the traffic is normal, i.e., no suspicious or malicious activities were introduced in the traffic flow. It is considered "normal" traffic and can pass without blocking.
(2) **C&C:** This tag denotes that the IoT device under test was communicating to a command-and-control server. This is detected in several ways; either there was periodic communication with the suspicious server, or the IoT device downloaded malicious binaries from this suspicious server.
(3) **DDoS**: This tag indicates that the IoT device was part of a distributed denial of service (DDoS) attack. This is detected by the number of flows targeting the same IP, which always come in a volumetric order.
(4) **Okiru:** This tag points to the behavior of the Okiru Botnet. This is detected when it matches the same pattern of the infamous Okiru traffic pattern.
(5) **Part of a horizontal port scan:** This tag shows that the traffic flow was used in the reconnaissance phase to gather more information about the target. This information is used in further attack steps by knowing the open ports, for example. This was detected by comparing the pattern of a similar port, in bytes, but having different destination IPs.

In order to obtain random sample data, we shuffled the rows 10 times so that the sample represented a complete random input for the model. This is close to the real traffic.

### 4.3. Selected Features

The dataset has a different type of information for each row (flow). Table 2 shows the different features that were captured for each data flow between the source and the

destination, regardless of it is normal or malicious traffic. This information is labeled in a separate column to be used later in the classification stages. We used two types of classification, binary classification and multiclass classification. Binary classification provides an output, whether the flow is malicious or normal traffic. This is important for the IDS in making a fast and prompt decision, as the traffic flows continuously in a real network between the EVCS and the EVCSMS. The second classification examined, the multiclass classification, measures the IDS system's effectiveness to detect the attack type. Both are important in the IDS's operation, but the first one is faster, as we see in the experiment results.

**Table 2.** Captured features in the IoT-23 dataset per flow.

| Feature | Description | Type |
|---|---|---|
| **ts** | The capture time, expressed in Unix Time | Integer |
| **uid** | Unique capture identification | String |
| **id_orig.h** | Source IP address | String |
| **id_orig.p** | Source port number | Integer |
| **id_resp.h** | Destination IP address IoT device | String |
| **id_resp.p** | Destination port number IoT device | Integer |
| **proto** | The type of network protocol (TCP/UDP) | String |
| **service** | Protocol used in the application | String |
| **duration** | The amount of time data traded between the IoT device and the attacker | Float |
| **orig_bytes** | The amount of data sent by the source to the destination device | Integer |
| **resp_bytes** | The amount of data sent by the destination IoT device back to the source | Integer |
| **conn_state** | The state of the connection | String |
| **local_orig** | Locally originated connection locally | Boolean |
| **local_resp** | Locally originated response | Boolean |
| **Missed bytes** | Number of missed bytes in the flow | Integer |
| **history** | The history of the connection state | String |
| **Tunnel parents** | The ID of the tunneled connection | String |
| **Label** | Capture type (malicious or normal) used in the binary classification | String |
| **Detailed label** | The type of malicious traffic used in the multiclass classification | String |

In this paper, we used a typical IoT dataset (IoT-23) with a typical IoT system, which is the EVCS system. We examined the different types of attacks, mainly the port scan, which is the first phase of the attack, and the DDoS, which is a typical destructive type of attack that can deem a system to be unavailable. In addition, the Okiru malware attack is examined as a clear showcase of the malware in IoT networks. Finally, the command and control (C&C) is the goal of the attacker to control the system remotely and give commands to execute everything the attacker wants to do on the target system. We used the summarized flows obtained from the original pcap files using the Zeek network analyzer.

From the above, we considered the EVCS as a practical example of an IoT system and applied the different machine learning algorithms using the IoT-23 dataset, which is more relevant to the problem under study.

As a preparation step for the data, we eliminated some of the features that are irrelevant to an actual attack. Some features were eliminated due to the weak correlation with the label, and other features were eliminated due to insufficient data, and hence do not take part in the decision. The following features have a very weak correlation with the label, as they are dependent on the conditions of the capture only. Therefore, to ensure that the decision was independent of weak features, we eliminated those features, i.e., **ts, uid, id.orig_h, and local_orig.h** to eradicate any dependency on the capturing environment.

Moreover, we eliminated the following features, as there were not many data; they contain a dash "-", which denotes that there is no information available. These features are **local_resp**, **missed_bytes**, and **tunnelparents**.

Table 3 presents the eliminated features that do not affect the output decision. We repeated the same experiment taking into consideration the full features, and we received the same results; this confirms that these features are not important in the decision.

**Table 3.** Eliminated features from the IoT-23 dataset.

| Feature | Elimination Reason |
| --- | --- |
| ts<br>uid<br>id.orig.h<br>local_orig | Weak correlation with the "label" |
| local_resp<br>missed_bytes<br>tunnel parents | No information, mostly dash "-" |

Finally, 14 out of 21 features were used in the experiment. Those features are id.orig_p, id.resp_p, proto, service, duration, orig_bytes, resp_bytes, conn_state, history, orig_pkts, orig_ip_bytes, resp_pkts, resp_ip_bytes, and label.

*4.4. Machine Learning Classifiers*

Using machine learning in securing the IoT [22] is a hot topic in academia. The selection of the best algorithm that matches the problem is a challenge. The target is to obtain an accurate result with the least processing power and, hence, a faster response time. This helps spot the attack in real time and make the decision to block the malicious traffic at once. Moreover, a principal factor here is that machine learning classifiers must be able to make the right decision when faced with limited data, i.e., trained with a limited amount of data and still have acceptable accuracy in the case of unknown attacks. We totally agree that unsupervised learning algorithms be widely used to deal with unknown attacks (zero-day attacks). However, one of the limitations of applying the unsupervised learning algorithms is that they suffer from high false alarms and low detection rates. To the best of our knowledge, the highest performance obtained by the unsupervised learning techniques does not exceed 90 percent, which is not acceptable in defending from unknown attacks. On the other side, implementing supervised learning algorithms can achieve high performance that can reach 99% or above. The achieved performance is similar to that of signature-based intrusion detection systems (such as snort). Now, back to the main question—what is the system response in case of a zero day attack? Supervised learning still can benefit from regularization techniques to simplify the system in order to overcome the overfitting problem [23] and, hence, supervised learning algorithms can have superior results over unsupervised learning of the same problem. Different regularization techniques, i.e., L1 and L2, have been used to address the problem of overfitting and to improve the capability of network-based IDSs in the detection of unseen intrusion events. For the sake of consistency, we have selected several classification algorithms and applied them to the same data sample. These algorithms are described in the next subsections.

(1)  Naïve Bayes classifier: Naïve Bayes (NB) may be a basic strategy for developing classifiers, represented as vectors of featured values, where the class labels are drawn from a few limited sets. There is no single calculation for preparing such classifiers, but rather a family of calculations based on a common rule: all naïve Bayes classifiers accept that the value of a specific feature is free of the value of any other feature, given the class variable. A naïve Bayes classifier considers each of these highlights to contribute freely to the likelihood of the feature, in case of any conceivable relationships

between the other features. An advantage of naïve Bayes is that it as it requires a small amount of training data to assess the parameters relevant for the classification.

(2)  J48 classifier: This algorithm is also called C4.5; it is categorized as a classification algorithm in which the output is a decision tree that is based on information theory [24]. It is not a new algorithm, as it is an extension of the ID3 algorithm by Ross Quinlan. It is named in WEKA tool (used in the simulation of this study) as J48, where J is for Java as it is an opensource implementation of C4.5. This algorithm is also known by statistical classifier, as its output is a decision tree based on the C4.5 algorithm. The difference between C4.5 and J48 is that J48 has more features than C4.5, such as decision tree pruning, accounting of missing values, etc.

(3)  The attribute-select classifier: The attribute-select classifier is a combination of two steps. The first step is dimensionality reduction (DR) through attribute selection; in this step, the dimensionality of the training and test data is reduced using attribute selection before being passed into the classifier, and the second step is classification. There is a variety of dimensionality reduction methods, but this is not the scope of this study. It chooses attributes based on the training data on the off chance that we are inside cross-validation. At that point, it trains the classifier, once more on the training data as it were, and after that, it assesses the entire classifier on the test data. This makes handling attribute selection totally straightforward, and the base classifier receives the diminished dataset.

(4)  Filtered classifier: This classifier is based on a basic idea of filtering out the irrelevant attributes and including only the relevant ones in the process [25]. This has a significant impact on both the processing time and efficiency of the algorithm. This is why the attribute selection algorithm is usually applied before the other tasks of classification, clustering, etc. Attribute selection is divided into two sequential parts. The first is subset generation, in which the searching process is used to compare between the determined subset and the candidate subset. If the candidate subset has better values, then it is labelled as the best one. This process is repeated several times until termination is reached. The second part is ranking, which is used to know the importance of the attributes. The ranking method can use statistics or can be based on the information theory [26,27].

### 4.5. Experimental Setup

The experiment was conducted using a machine with the operating system Windows 11 Pro 64-bit version 21H2, build 22000.739. The processor was an Intel® Core™ i7-1165G7, CPU @ 2.8 GHz (8 Cores). The memory of the machine was 16 GB. The machine was equipped with a graphics card, Intel® IRIS® Xe graphics with 4 GB RAM. The tool used for the simulation was Waikato Environment for Knowledge Analysis (WEKA) version 3.8.5 64-bit. This is illustrated in Table 4.

**Table 4.** Hardware and software specifications used in the simulation.

| Operating System | Windows 11 Pro 64-bit Version 21H2, 22000.739 Build |
|---|---|
| Processor | Intel® Core™ i7-1165G7, CPU @ 2.8 GHz (8 Cores) |
| Memory | 16 GB DDR3 |
| Graphics card | Intel® IRIS® Xe graphics with 4GB RAM |
| Software | WEKA version 3.8.5 64-bit |

As mentioned earlier in the previous section, the 21 features were reduced into 14 features after eliminating seven unnecessary features, in which the new reduced dataset, containing the 14 unique features, was the model's input. Two sets of experiments were conducted, the first was for binary classification, and the second was for multiclass classification. The next section demonstrates the results for each classification.

## 5. Simulation

In this section, we present the experimental results. The evaluation metrics are proposed in Section 5.1, while experimental results are presented in Section 5.2.

### 5.1. Evaluation Metrics

The metrics described below were used to fairly evaluate the algorithms. These results are discussed and concluded in the discussion and limitations section. The arguments used in these metrics were first presented in a previous study [25]. Other algorithms proposed in the literature [14,15] used the same evaluation metrics.

TP, TN, FP, and FN denote true positives, true negatives, false positives, and false negatives.

- TP: amount of perfectly identified malicious traffic.
- TN: amount of perfectly detected non-malicious traffic.
- FP: amount of wrongly detected malicious traffic as positive that is non-malicious.
- FN: amount of wrongly detected non-malicious traffic that is malicious.

#### 5.1.1. Confusion Matrix

A confusion matrix is a table that allows for the visualization of the performance of a model by showing which values the model thought to belong to which classes. It has an $N \times N$ size, where $N$ is the number of classes, with the columns representing the actual classes and the predicted classes' rows.

#### 5.1.2. Precision

Precision is the metric that evaluates the model by calculating the fraction of correctly identified positives.

It is calculated using the following formula:

$$Precision = \frac{TP}{TP + FP} \tag{1}$$

#### 5.1.3. Accuracy

The accuracy metric evaluates the model by calculating the fraction of correct predictions over the total number of predictions.

It is calculated using the following formula:

$$Accuracy = \frac{TP + TN}{TP + TN + FP + FN} \tag{2}$$

#### 5.1.4. Recall Score

Recall score is a metric that evaluates the model by calculating the fraction of actual positives that was correctly identified.

It is calculated using the following formula:

$$Recall = \frac{TP}{TP + FN} \tag{3}$$

#### 5.1.5. F-1 Score

F-1 score is a metric that calculates the harmonic mean of the precision and the recall score. It is considered an extension of the accuracy metric, as it considers both false positives and false negatives in the same equation.

It is calculated using the following formula:

$$F_1 = 2 * \frac{Precision * Recall}{Precision + Recall} \tag{4}$$

While the F-1 score is originally used for binary classification, it can also be used in multiclass classification by macro-averaging, where all the classes are considered equally. The result in this paper is based on macro-averaging.

$$F_1(Macro) = \sum_{i=1}^{n} \left[ F_1(i) * \frac{1}{n} \right] \qquad (5)$$

*5.2. Experimental Results*

ML algorithms address application problems by using a dataset for learning. The dataset is divided into training and testing sets. The training set is used to learn and study the dataset's various features. For instance, given an intrusion detection dataset, the algorithms learn features from the training dataset to classify a given sample as an attack or normal. The ML algorithm's task is to improve the algorithm's classification accuracy by performing behavioral analysis of normal and attack traffic scenarios in the network. ML algorithms are categorized into classification and clustering algorithms. Classification algorithms work with labeled data samples and build prediction models by analyzing input parameters and mapping them with the expected output. Thus, these methods build the relationship between input and output parameters. In the training phase of the classification algorithm, the learning model is trained using a training set. The learning in the training phase is then utilized to predict and classify new data input.

Classification methods are recognized for learning using data representation and labeling, whereas clustering methods are known for learning using unlabeled datasets. The result for the simulation is split into binary classification results and multiclass classifications results.

### 5.2.1. Binary Classification Results

For the binary classification results, we evaluated the algorithm's performance with 285,000 flows, with normal/malicious traffic labels equivalent to 73,000/212,000. First, we evaluated the time spent building the model using the hardware specifications in the experimental setup. This time varies from one hardware to another, but all the experiments were carried out on the same hardware, so the values are used for comparison. In addition, the accuracy values were presented. We then evaluated the precision, recall, and F-1 score.

The results for the response time and the accuracy of the four algorithms in the binary classification are illustrated in Figure 2.

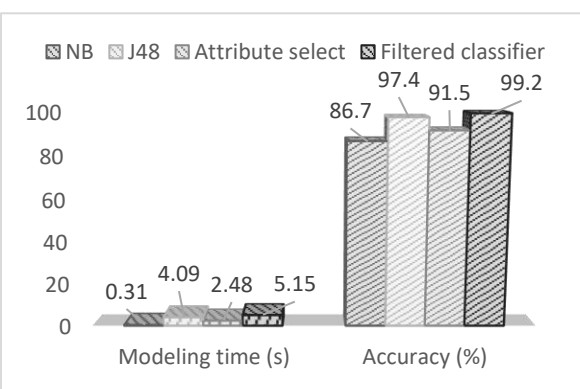

**Figure 2.** Response time versus accuracy for binary classification.

The results for the binary classification in Table 5 show superior results in terms of accuracy for the J48 classifier (97.4%) and the filtered classifier (99.2%). In comparison, the lowest accuracy was obtained from the naïve Bayes algorithm (86.7%). We also noticed that the filtered classifier consumed more time (5.15 s) in the modeling than the J48 classifier (4.09 s).

**Table 5.** Binary classification simulation results.

|  | Response (s) | Accuracy (%) |
|---|---|---|
| **Naïve Bayes** | 0.31 | 86.7 |
| **J48 classifier** | 4.09 | 97.4 |
| **Attribute select** | 2.48 | 91.5 |
| **Filtered classifier** | 1.15 | 99.2 |

However, in terms of precision, recall, and F-1 score, the filtered classifier achieved the highest rank among all other algorithms, as shown in Table 6.

**Table 6.** Precision, recall, and F-1 score for binary classification.

| Model | Precision (%) | | Recall (%) | | F-1 Score (%) | |
|---|---|---|---|---|---|---|
|  | **Normal** | **Attack** | **Normal** | **Attack** | **Normal** | **Attack** |
| **Naïve Bayes** | 0.84 | 0.87 | 0.59 | 0.96 | 0.69 | 0.91 |
| **J48** | 0.92 | 0.99 | 0.98 | 0.97 | 0.95 | 0.98 |
| **Attribute select** | 0.76 | 0.98 | 0.97 | 0.89 | 0.85 | 0.94 |
| **Filtered classifier** | 0.97 | 1 | 0.99 | 0.99 | 0.98 | 0.99 |

Figure 2 depicts the binary classification results, which show the response time of the model in seconds and the accuracy in percentage for the four algorithms under test.

The filtered classifier achieved the highest results in terms of precision, recall, and F-1 score.

### 5.2.2. Multiclass Classification Results

For the multiclass classification results, we evaluated the algorithm's performance with 285,000 flows, with a detailed label of the type of attack. The number of samples per class is illustrated in Table 7. The same was performed during the binary classification. First, we evaluated the response time of the algorithm and the accuracy; then, we evaluated the precision, recall, and F-1 score.

**Table 7.** Number of samples per class.

| Class | Number of Samples |
|---|---|
| Benign | 73,082 |
| C&C | 15,044 |
| DDoS | 51,361 |
| Okiru | 78,603 |
| Part of a horizontal port scan | 67,849 |

The results for the response time and the accuracy for the four classifier algorithms under test in the multiclass classification are presented in Table 8.

Figure 3 depicts the multiclass classification results, which show the response time of the model in seconds and the accuracy in percentage for the four classifier algorithms under test.

**Table 8.** Multiclass classification simulation results.

|  | Response (s) | Accuracy (%) |
|---|---|---|
| Naïve Bayes | 0.3 | 77 |
| J48 classifier | 4.23 | 99.2 |
| Attribute select | 2.81 | 97.19 |
| Filtered classifier | 1.62 | 99.2 |

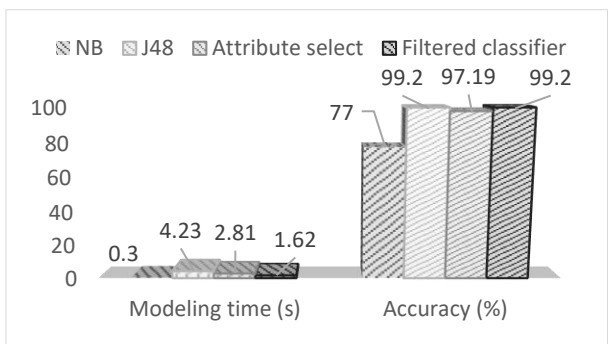

**Figure 3.** Response time versus accuracy for multiclass classification.

From the results of the multiclass algorithms, again as the binary classification, the J48 classifier and the filtered classifier both achieved the highest accuracy (99% for both). In contrast to the binary classification, the modeling time for the filtered classifier (1.62s) was lower, with a more significant value than the J48 classifier (4.23s). The reason behind the quicker modeling time for the filtered classifier in the multiclass (1.62 s) than the binary classification (5.15 s) is due to the nature of the filtered classifier algorithm. It evaluates each class separately and in parallel (five small chunks of parallel operations). This is as opposed to the binary classification, which deals with the whole data set (285,000 samples) at once. Moreover, the naïve Bayes algorithm achieved the lowest accuracy result (77%). In terms of precision, recall, and F-1 Score, both the J48 classifier and the filtered classifier achieved the same results, as shown in Table 9.

**Table 9.** Precision, recall, and F-1 score for multiclass classification.

| Model | Precision (%) | | Recall (%) | | F-1 score (%) | |
|---|---|---|---|---|---|---|
|  | Normal | Attack | Normal | Attack | Normal | Attack |
| **Naïve Bayes** | 0.99 | 0.85 | 0.59 | 0.77 | 0.59 | 0.77 |
| **J48** | 0.97 | 0.99 | 0.99 | 0.99 | 0.98 | 0.99 |
| **Attribute select** | 0.91 | 0.97 | 0.98 | 0.97 | 0.94 | 0.97 |
| **Filtered classifier** | 0.97 | 0.99 | 0.99 | 0.99 | 0.98 | 0.99 |

Notice that the figures for the precision, recall, and F-1 are the weighted average among the classes in the dataset.

Comparing the results with other state-of-the-art methods, as in other studies [19,20], shows the superior results of using the filtered classifier (accuracy of 99.2%) over the other deep learning algorithms (maximum accuracy of 97%).

Table 10 represents the accuracy in detecting different types of attacks using the filtered classifier.

**Table 10.** Accuracy of the filtered classifiers for different types of attacks.

| Attack | Description | Result | Accuracy |
|--------|-------------|--------|----------|
| **Benign** | That is the normal traffic | Detected | 99.9% |
| **DDoS** | Denial of service | Detected | 100% |
| **C&C** | Communication with command-and-control server | Detected | 99.8% |
| **Botnet** | Botnet traffic used to employ the IoT device in other attacks—Okiru | Detected | 100% |
| **Scan** | Part of the horizontal scan is the initial phase of reconnaissance that the attacker uses to learn the open ports | Detected | 96.8% |
| Overall | Weighted average | | 99.2% |

The model successfully classified the traffic as either benign or malicious (in the binary classification). It classified the different types of attacks (in the multiclass classification) with a very high accuracy of 99.2% using the filtered classifier algorithm.

## 6. Discussions and Limitations

The issue of security in the EVCS is a real industrial problem. The effect of a cyberattack on the EVCS can be catastrophic if exploited by malicious actors and state-sponsored attack groups. With the limited number of EVCSs currently deployed and the increasing number of electric vehicles with short battery range, any downtime in one EVCS can affect the travel plans for many EV users. Furthermore, the whole power grid can be halted due to a cyberattack, which can directly affect the economy. If we also add unknown exploits, known as zero days, the risk is even higher. In order to mitigate this risk, there a need for an accurate and efficient algorithm. The use of machine learning to build an IDS engine was discussed in this paper. In order to evaluate the proposed IDS correctly, a relevant dataset is also needed to represent the real traffic and attacks. In this paper, we used the IoT-23 dataset, which is built from native IoT network traffic, to evaluate four different machine learning classifier algorithms that can be used in ML-based IDS. Each classifier has different logic behind its theory of operation. We noticed, from the results, that filtered classifiers perform very well on testing data in terms of accuracy and other metrics. Hence, they can be applied for zero-day attack detection.

Likewise, our study experienced some limitations, which are listed below:

- Although deep learning (DL) is widely used in several application domains, such as image prepossessing, language translation, etc., using DL is out of the scope of this paper.
- We trained and evaluated the ML algorithms in offline mode using virtual simulation without implementing a physical EVCS system. However, the detection of attacks online is very important to understanding how this IDS can handle intrusion in real time.
- For intrinsic evaluation, we should consider several datasets in our comparison. However, in this work, we only used the IoT-23 dataset to train and evaluate different ML algorithms. We are planning to build our own dataset from a real EVCS system and consider several datasets to build a robust intrusion detection model.

## 7. Conclusions and Future Work

The EVCS ecosystem is vulnerable to many types of attacks targeting IoT systems. Therefore, the need for an accurate and efficient method of detection arises. In this paper, the proposed ML-IDS in IoT EVCS was used to detect the different types of attacks by malicious actors. We surveyed other research efforts in the literature that have touched on the same subject. We selected the latest IoT dataset and used it to examine four different types of classical machine learning classifiers. We found that the filtered classifier is the best choice for anomaly detection and classification using the IoT23 dataset, as it achieved the highest accuracy in both binary classification and multiclass classification and also in terms of precision, recall, and F-1 score. The result of this study is in line with other

studies as well. The proposed algorithm can also be applied to any critical industrial control system (ICS), such as SCADA systems and green hydrogen control systems, to enhance their security resilience as they were originally built without taking security into consideration. We believe this paper's findings will help build a comprehensive IDS by recognizing that classification models should be trained with the relevant dataset addressing the relevant application. It is highly recommended to build a dedicated dataset for EVCSs that will benefit the researchers in developing and examining different types of attacks. Future research aims to find the minimum amount of data to be used in training while preserving the same accuracy level. Future studies should also focus on measuring the impact of feature selection and consider new methodological steps to developing deep learning models. For future work, we will use deep learning algorithms to evaluate system performance with different datasets.

**Author Contributions:** Conceptualization, M.E.; methodology, M.E.; formal analysis, M.E., M.S.E. and A.D.J.; investigation, M.S.E.; writing—original draft preparation, M.E.; visualization, M.S.E. and M.A.; supervision, M.S.E., A.D.J. and M.A.; project administration, M.A.; funding acquisition, A.D.J. All authors have read and agreed to the published version of the manuscript.

**Funding:** This research received no external funding.

**Conflicts of Interest:** The authors declare no conflict of interest.

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
