# Peer review of "A Machine Learning-Based Intrusion Detection System for IoT Electric Vehicle Charging Stations (EVCSs)"

_electronics, doi:10.3390/electronics12041044_

Round 1
Reviewer 1 Report (New Reviewer)
The manuscript entitled "Machine Learning-based Intrusion Detection System for IoT Electric Vehicle Charging Station (EVCS) “is a good work on the Computer Science. However, I have some minor concerns about the writing part of the article. Hence, after a minor modification of the article it can be considered for publication.
1. The abstract seems to be very short; I suggest to follow
2. Author should show a summary from the twenty malicious scenarios in the form of table.
3. I suggest to include recent articles in reference portion.
4. Figure quality need to improve.
Author Response
The authors would like to extend their appreciation to the area editor and the reviewers for their precious time and invaluable comments.
Only one thing to be noticed, that reviewer #1 is refereeing to a different paper ID, since our Manuscript ID: electronics-2184464, entitled "A Machine Learning-based Intrusion Detection System for IoT Electric Vehicle Charging Station (EVCS)".
Reviewer 2 Report (New Reviewer)
A Machine Learning-based Intrusion Detection System for IoT Electric Vehicle Charging Station (EVCS)
This paper proposes a classifier algorithm for detecting malicious traffic in the IoT environment using Machine Learning. The proposed system uses a real IoT dataset derived from real IoT traffic. Multiple classifying algorithms are evaluated. Results were obtained on both binary and multiclass traffic models. Using the proposed algorithm in the IoT-based IDS engine serving the Eclectic Vehicle charging stations will bring stability and eliminate a substantial number of cyberattacks that may disturb day-to-day life activities.
Some of the suggestions needs to be addressed before acceptance
In introduction the motivation and methodology of the work must be clear
Figure 2: Methodology in building the model is not informative
As a preparation step for the data,the authors said they have eliminated some of the features that are irrelevant to an actual attack. Some features were eliminated due to the weak correlation with the label, and other features were eliminated due to insufficient data. The authors should explain what are the features used finally.
Why did they eliminated those features i.e.ts , uid , id.orig_h, and local_orig.h to erradicate any dependency on the capturing environment. What are the effects on the results.
The authors can refer
Autonomous Vehicles in 5G and beyond: A Survey
Green-Tech CAV: Next Generation Computing for Traffic Sign and Obstacle Detection in Connected and Autonomous Vehicles
Author Response
Reviewer#2, Concern #1:
- In introduction the motivation and methodology of the work must be clear.
Authors response: We thank the reviewer for the time given in the review of our manuscript. The raised comment for the motivation and methodology were carefully addressed.
Authors action: Added a paragraph in the introduction to focus on the motivation and methodology of the work.
Added section
“To address these challenges, the use of machine learning (ML) techniques has become increasingly important in detecting and preventing cyberattacks on EVCS. ML algorithms are well suited for this task due to their ability to learn from large amounts of data and make predictions based on that learning. With the increasing use of EVs and EVCS, there is a growing need for effective and efficient methods to detect and prevent cyberattacks
In this paper, we propose the use of ML algorithms for detecting cyberattacks on EVCS. The methodology involves collecting and pre-processing data from various sources including the charging station’s network traffic, the charging process, and the charging station’s environment. This data is then used to train and evaluate various ML models for detecting and classifying potential cyberattacks. Our study will compare the performance of different ML algorithms and provide insights into the most effective methods for detecting cyberattacks on EVCS”
Reviewer#2, Concern #2:
- Figure 2: Methodology in building the model is not informative
Authors response: The authors agree that that is the figure could have been more informative, but it represents the standard methodology for all the experiments and doesn’t add new to the reader.
Authors action: The Authors decided to remove the figure and add more explanation in the text.
Reviewer#2, Concern #3:
3- As a preparation step for the data, the authors said they have eliminated some of the features that are irrelevant to an actual attack. Some features were eliminated due to the weak correlation with the label, and other features were eliminated due to insufficient data. The authors should explain what are the features used finally.. Why did they eliminated those features i.e.ts , uid , id.orig_h, and local_orig.h to erradicate any dependency on the capturing environment. What are the effects on the results.
Authors response: We’d like to thank the reviewer for this suggestion and agree that we should list the features that were used in the final experiment.
For the Query of why the features were eliminated, we’ve added the explanation “ the eliminated features that do not affect the output decision. We repeated the same experiment taking into consideration the full features, and we received the same results, this confirms that these features are not important in the decision”
Authors action: We’ve added the used features in the experiment after the elimination.
Added section: “Finally, 14 features out of 21 features were used in the Experiment. Those features are id.orig_p, id.resp_p, proto, service, duration, orig_bytes, resp_bytes, conn_state, history, orig_pkts, orig_ip_bytes, resp_pkts, resp_ip_bytes, and Label.”
Reviewer#2, Concern #4:
4- The authors can refer
Autonomous Vehicles in 5G and beyond: A Survey
Green-Tech CAV: Next Generation Computing for Traffic Sign and Obstacle Detection in Connected and Autonomous Vehicles
Authors response: The Authors would like to thank the reviewer for the suggestion.
Authors action: Authors added those valuable papers in the reference section
[2] Hakak S, Gadekallu TR, Maddikunta PK, Ramu SP, Parimala M, De Alwis C, Liyanage M. Autonomous Vehicles in 5G and beyond: A Survey. Vehicular Communications. 2022 Nov 28:100551.
[3] Kumar VA, Raghuraman M, Kumar A, Rashid M, Hakak S, Reddy MP. Green-Tech CAV: Next Generation Computing for Traffic Sign and Obstacle Detection in Connected and Autonomous Vehicles. IEEE Transactions on Green Communications and Networking. 2022 Mar 28;6(3):1307-15.
Reviewer 3 Report (Previous Reviewer 2)
The authors have addressed all my previous concerns, and the manuscript is acceptable in its current form.
Author Response
Once again, we thank you for the time you spent reviewing our paper, and was greatly appreciated for accepting our article.
This manuscript is a resubmission of an earlier submission. The following is a list of the peer review reports and author responses from that submission.
Round 1
Reviewer 1 Report
I'm going to recommend this article for rejection - the main reason is that the introduction and motivation for the work waxes lyrical about electric vehicle charging systems (EVCS), but then the experiments in the paper are fairly standard machine learning algorithms applied to a dataset that contains a few raspberry pis acting as malicious agents, some Phillips smart globes, a smart doorlock and an Amazon echo speaker. No EVCS in sight!
Also, the underscore the importance of considering these systems, the paper describes a denial of service attack on the electricity grid by coordinating compromised EVCSes to overload the grid by alternating synchronous charge/discharge cycles. A problem indeed, which is already being addressed as it is also becoming an issue as more intermittent renewable sources are brought online, without the need for malicious actors. The biggest issue is that I fail to see how the work reported in the paper even addresses that issue, so why mention it?
Of course, there are multiple other issues with the paper, including poor English expression and copy editing, and also it not being clear whether the full (pcap) dataset of IOT23 was being used, or just the flow summaries that have already been preprocessed. But these are secondary to the main issue that the research work doesn't even address the motivations set out in the paper's introduction.
Author Response
Reviewer#1, Concern #1:
- I'm going to recommend this article for rejection -The main reason is that the introduction and motivation for the work waxes lyrical about electric vehicle charging systems (EVCS), but then the experiments in the paper are fairly standard machine learning algorithms applied to a dataset that contains a few raspberry pis acting as malicious agents, some Phillips smart globes, a smart doorlock and an Amazon echo speaker. No EVCS in sight!
Authors response: We appreciate the reviewer’s time during the review and would like to clarify a fundamental point here. We are dealing with the EVCS ecosystem as an abstract IoT system. Actually, we are dealing with the IoT components of the EVCS. Furthermore, we explained the components of the EVCS and how the Intrusion detection system (IDS) will defend it from the IoT attacks. We proposed usage of an effective classification algorithm in the IDS that is sniffing the traffic of the EVCS (as an IoT system). The superior results obtained in the experimental results, can protect the EVCS system from different types of IoT attacks targeting the EVCS.
In other words, the EVCS is in the application layer, while the actual underlying layer serving the EVCS is a pure IoT devices communicating with each other’s, which is the target of the study. This means also that the results of this study can be applied to any IoT system. This makes the study more powerful than limiting it to only one type of application.
Authors action: We added more clarification in the Introduction section.
Reviewer#1, Concern #2:
2- Also, the underscore the importance of considering these systems, the paper describes a denial of service attack on the electricity grid by coordinating compromised EVCSes to overload the grid by alternating synchronous charge/discharge cycles. A problem indeed, which is already being addressed as it is also becoming an issue as more intermittent renewable sources are brought online, without the need for malicious actors. The biggest issue is that I fail to see how the work reported in the paper even addresses that issue, so why mention it?
Authors response: Thanks for the detailed review of the paper and the punctual review. Regarding the raised concern, mentioning the probable consequences of cyber-attacks is an important factor for the decision makers in qualifying and quantifying the risks. If the system is unprotected and the attacker can gain access to the system, and can deliberately cause the problem of alternating charge/discharge for making a wide effect of the attack. This is a real risk that need to be highlighted.
The motivation behind mentioning this problem is to emphasize the importance of having an effective and accurate intrusion detection system that can defend the system from malicious actors, who can cripple the system if they were able to gain access by one of the mentioned attacks in the study.
Authors action: We updated the manuscript by adding more clarification regarding this point.
Reviewer#1, Concern #3:
3- Of course, there are multiple other issues with the paper, including poor English expression and copy editing, and also it not being clear whether the full (pcap) dataset of IOT23 was being used, or just the flow summaries that have already been preprocessed. But these are secondary to the main issue that the research work doesn't even address the motivations set out in the paper's introduction
Authors response: We thank the reviewer for the detailed review. Regarding the English expression and copy editing, the paper was reviewed again, and many sentences were amended. Also, extensive editing was done to ensure proper punctuation. Regarding the data used from the IoT dataset, the used data was the flow summaries obtained by the Zeek network analyzer, which have been preprocessed.
Authors action: We reviewed and proofread the manuscript addressing the English expressions and punctuation. Also added clarification about the preprocessing of the data in section “4.3 feature selection”
Reviewer 2 Report
The work projects a system for detecting malicious traffic in the IoT environment using Machine Learning, and the proposed system uses a real IoT dataset derived from real IoT traffic. This work posed a significant impact in the domain and will benefit the readers. However, I have a major concern that should be addressed by the authors before publication.
1. The performance analysis and security examinations lack the required rigors. We need to see how the projected model is subjected to several security vulnerability tests. The mathematical proofs and validation criteria need to be well articulated.
2. There is a need to compare the projected system with the related state-of-the-art to demonstrate its efficacy and superiority over the existing schemes.
3. The tables are not formatted according to the Journal style, and moderate English language editing of the paper is requested.
4. The following reference material could be useful to the current work: https://content.iospress.com/articles/journal-of-intelligent-and-fuzzy-systems/ifs220310
Author Response
Reviewer#2, Concern #1:
- The performance analysis and security examinations lack the required rigors. We need to see how the projected model is subjected to several security vulnerability tests. The mathematical proofs and validation criteria need to be well articulated.
Authors response: We agree on reviewer comment. However different types of attacks that the model is subject to, are already mentioned in section 4.2 “IoT-23 Dataset” as the traffic injected is typically a collection of different kind of attacks
Authors action: We updated the manuscript by adding a table summarizing the different types of attacks in the results section 5.2.2 Multiclass classification results - Table 10.
Regarding the mathematical proofs and Validation criteria, we updated section “5.1 Evaluation metrics” with more clarification.
Reviewer#2, Concern #2:
2- There is a need to compare the projected system with the related state-of-the-art to demonstrate its efficacy and superiority over the existing schemes.
Authors response: We agree on reviewer comment. Comparing the results with other state-of-the-art will enrich the study and will show the superiority of using this model over another models. However, the study itself is comparing 4 different classifying algorithms (Naïve bayes, J48 Classifier, Attribute select classifier and the Filtered classifier. The Filter classifier is showing superior results compared to the 3 other classifiers.
Authors action: We have updated the manuscript by adding a paragraph in the simulation results section
Reviewer#2, Concern #3:
- The tables are not formatted according to the Journal style, and moderate English language editing of the paper is requested.
Authors response: The authors would like to thank the reviewer for his careful review.
Author action: The manuscript was proofread, tables were formatted to the Journal style and more language and punctuation editing was performed.
Reviewer#2, Concern #4:
- The following reference material could be useful to the current work: https://content.iospress.com/articles/journal-of-intelligent-and-fuzzy-systems/ifs220310
Author response: Valuable resource, was cited in the paper and added to the references.
Reviewer 3 Report
The authors proposed a system for detecting malicious traffic in the IoT environment using Machine Learning. The proposed system uses a real IoT dataset derived from real IoT traffic. Multiple classifying algorithms are evaluated. Results are obtained on both binary and multiclass traffic models. Using the proposed algorithm in the IoT-based IDS engine serving the Eclectic Vehicle charging stations (EVCS) will bring stability and eliminate a substantial number of cyberattacks that may lead to the disturbance of day-to-day life activities.
There are some issues which are required to be incorporated in the manuscript.
1. The results are not compared with the existing related techniques.
2. Tables number 5 and 7 are repeated. Change the numbers of the tables.
3. Include the correlations values between the output feature and the dropped features of the dataset. Also check the name of the dropped features as there is some mismatch in the feature names in Table number 2 and 3.
4. Some of the abbreviations are repeated.
5. In Introduction section, in sentence-"All these components of the EVCS ecosystem are subject to the different kinds of IoT cyber-attacks", about which components the authors are talking.
6. Kindly use small sentences, for example- . That is to be able to offer extended services to the customers, but this will enable a set of cyber-attacks against the whole EVCS ecosystem. the effect is not limited to the EVCS only, but it is extended to the critical infrastructure of the power grid and the end users equally- is a very large sentence. Break it into atleast 2 sentences.
7. In second line of Section 2, sponsors actors should be replaced by sponsor actors.
8. Use single work- either system or technique or algorithm throughout the paper.
Author Response
Reviewer#3, Concern #1:
1- The results are not compared with the existing related techniques
Authors response: We agree on reviewer comment. Comparing the results with other state-of-the-art will enrich the study and will show the superiority of using this model over another models. However, in section 4.4, the study itself compares 4 different classifying algorithms (Naïve bayes, J48 Classifier, Attribute select classifier and the Filtered classifier. The Filter classifier shows superior results compared to the 3 other classifiers.
Author action: We updated the manuscript by adding more clarifications in the simulation results section. The clarifications compare between the state-of-the-art results compared to our results.
Reviewer#3, Concern #2:
2- Tables number 5 and 7 are repeated. Change the numbers of the tables.
Author response: The Authors agree on the reviewer comment.
Authors action: We updated the manuscript by the correct table numbers from table 6 till table 9.
Reviewer#3, Concern #3:
3- Include the correlations values between the output feature and the dropped features of the dataset. Also check the name of the dropped features as there is some mismatch in the feature names in Table number 2 and 3.
Author response: The Authors agree on the reviewer’s comment. New updates were added to further clarify the used features of the dataset.
Author action: Added more clarification on the section of “4.3 selected features” to be clearer to the reader. Also mismatch between table 2 and 3 were rectified.
Reviewer#3, Concern #4:
4- Some of the abbreviations are repeated.
Author action: we updated the manuscript and removed the duplicated abbreviations so that any abbreviation is mentioned once.
Reviewer#3, Concern #5:
5- In Introduction section, in sentence-"All these components of the EVCS ecosystem are subject to the different kinds of IoT cyber-attacks", about which components the authors are talking.
Authors response: The authors agree with the comment and added a sentence which clarifies the EVCS components which consist of the charging station, the power grid and the end users.
Author action: Updated the Introduction section to clarify the components.
Reviewer#3, Concern #6:
6- Kindly use small sentences, for example- . That is to be able to offer extended services to the customers, but this will enable a set of cyber-attacks against the whole EVCS ecosystem. the effect is not limited to the EVCS only, but it is extended to the critical infrastructure of the power grid and the end users equally- is a very large sentence. Break it into at least 2 sentences.
Authors response: We thank the reviewer for the careful revision. We agree on the reviewer’s comment, that the large sentences need to be broken.
Author action: The mentioned sentences Broken into small sentences. We revised and updated the manuscript for similar large sentences.
Reviewer#3, Concern #7:
7-In second line of Section 2, sponsors actors should be replaced by sponsor actors.
Author action: Changed to “state sponsor actors”
Reviewer#3, Concern #8:
8- Use single work- either system or technique or algorithm throughout the paper.
Author action: We unified the terms to use the term “algorithm” throughout the paper in describing the used classifier algorithm. While we used the term “system” in describing the whole EVCS system and/or the IoT system. We’ve omitted the term technique to avoid confusion.
Round 2
Reviewer 1 Report
Due to time constraints applied by the editors, I have only given this revision a cursory look over.
Minor, but right up front: "The Demand for Electric Vehicles (EVs) is growing exponentially." This is basically a meaningless sentence. Having my money in an account earning 0.1% interest (like was common recently) is growing exponentially. But it certainly won't make me rich any time soon. I know this expression is abused in the popular press, but this is a technical paper, and the you should know better. Perhaps replace "exponentially" with "rapidly" or "dramatically" depending on what you mean.
But more importantly, I don't think the connection between the discussion on EVCS and the technical work applying ANNs to a particular reference dataset is appropriately made. It is not stated that the reference dataset doesn't contain any data from EVCSes for example. Also for intrusion detection systems, there is the problem of "unknown unknowns". Supervised training is really the wrong way to go. We don't know ahead of time what sort of attacks will be carried out, so the problem is one of unsupervised training of anomaly detection.
Maybe this paper could be split into two: a discussion paper on the security issues of EVCSes, and a second paper on applying supervised learning of RNNs to the IOT-23 dataset. I wouldn't review the former paper, it is rather outside my bailiwick, but possibly I could review the latter. But I can't accept the paper in its current form.
Author Response
Reviewer#1, Concern #1:
- Due to time constraints applied by the editors, I have only given this revision a cursory look over.
Authors response: We thank the reviewer for the time given in the review of our manuscript. The newly raised comments in this round of review were carefully addressed.
Reviewer#1, Concern #2:
- Minor, but right up front: "The Demand for Electric Vehicles (EVs) is growing exponentially." This is basically a meaningless sentence. Having my money in an account earning 0.1% interest (like was common recently) is growing exponentially. But it certainly won't make me rich any time soon. I know this expression is abused in the popular press, but this is a technical paper, and the you should know better. Perhaps replace "exponentially" with "rapidly" or "dramatically" depending on what you mean.
Authors response: The authors see that term “exponentially” is suitable term to be used. If we observe the number of EVs in one country, we will find that it starts with a limited number, and by time, it increases by an exponential rate as more users start to use it. Many EV manufacturers join the market to provide more car models in parallel once the adoption increases, resulting in a burst increase in the number of EVs in an exponential way (Norway and Sweden is an example). However, we agree with the reviewer that “rapidly” can serve the same purpose.
Authors action: We appreciate the reviewer comment, and we changed the expression to “rapidly”.
Reviewer#1, Concern #3:
3- But more importantly, I don't think the connection between the discussion on EVCS and the technical work applying ANNs to a particular reference dataset is appropriately made. It is not stated that the reference dataset doesn't contain any data from EVCSes for example. Also for intrusion detection systems, there is the problem of "unknown unknowns". Supervised training is really the wrong way to go. We don't know ahead of time what sort of attacks will be carried out, so the problem is one of unsupervised training of anomaly detection.
Authors response: Unfortunately, there is no dedicated dataset for the EVCS till the moment, and that is a limitation in literature, which was one of the recommendations in the future work. However, we deal with the EVCS as an IoT component, thus we believe that the IoT-23 dataset is the most relevant dataset to the application of the EVCS. As there is no big difference between the traffic structure of the EVCS and the IoT.
Regarding the comment of the “unknown unknowns”, we would like to thank the reviewer for this comment as it touches the core of the paper. We totally agree that the unsupervised learning algorithms are widely used to deal with unknown attacks (zero-day attacks). However, one of the limitations of applying the unsupervised learning algorithms is that they suffer from high false alarms and low detection rates. To the best our knowledge, the highest performance obtained by the unsupervised learning techniques doesn’t exceed 90 percent, which is not acceptable in defending from unknown attacks. On the other side, Implementing the supervised learning algorithms can achieve high performance that can reach 99% or above. The achieved performance is similar to that of the signature-based intrusion detection systems (like snort). Now, back to the main question that - what will be the system response in case of a zero day attack? The supervised learning still can benefit from the regularization techniques to simplify the system to overcome the overfitting problem and hence the supervised learning algorithms can have superior results over the unsupervised learning for the same problem. Different regularization techniques i.e. L1 and L2 have been used to address the problem of overfitting and to improve the capability of Network based IDSs in detection of unseen intrusion events. Besides, supervised learning techniques have been widely used for Intrusion detection in the literature – to name a few
- Adamu, U.; Awan, I. Ransomware prediction using supervised learning algorithms. In Proceedings of the 2019 7th International Conference on Future Internet of Things and Cloud (FiCloud), Istanbul, Turkey, 26–28 August 2019; pp. 57–63
- ElSayed, Mahmoud Said, et al. "A novel hybrid model for intrusion detection systems in SDNs based on CNN and a new regularization technique." Journal of Network and Computer Applications 191 (2021): 103160.
- Urooj, Umara, et al. "Ransomware detection using the dynamic analysis and machine learning: A survey and research directions." Applied Sciences 12.1 (2021): 172.
- Said Elsayed, Mahmoud, et al. "Network anomaly detection using LSTM based autoencoder." Proceedings of the 16th ACM Symposium on QoS and Security for Wireless and Mobile Networks. 2020.
- V. Amanoul and A. M. Abdulazeez, “Intrusion detection system based on machine learning algorithms: A review,” in 2022 IEEE 18th International Colloquium on Signal Processing & Applications (CSPA). IEEE, 2022, pp. 79–84
- Ahmad, I. Alsmadi, W. Alhamdani, and L. Tawalbeh, “A comprehensive deep learning benchmark for iot ids,” Computers & Security, vol. 114, p. 102588, 2022.
Authors action: We added more clarification in section 4.4 for the usage of supervised learning over the unsupervised learning and the regularization techniques.
Reviewer#1, Concern #4:
4- Maybe this paper could be split into two: a discussion paper on the security issues of EVCSes, and a second paper on applying supervised learning of RNNs to the IOT-23 dataset. I wouldn't review the former paper, it is rather outside my bailiwick, but possibly I could review the latter. But I can't accept the paper in its current form
Authors response: The application of the EVCS and the underlying layer of using the ML in detecting the intrusions targeting the EVCS can’t be decoupled at the moment as the reader will miss the big picture and the contribution to the literature will be less.
Authors action: We believe that the paper structure is designed such that the sections are following a logic flow. We are afraid that splitting the paper will significantly affect the paper’s quality. In the near future, we plan to have different publications focusing on each subject.
Reviewer 2 Report
The authors have not addressed an earlier comment satisfactorily.
- The performance analysis and security examinations lack the required rigors. The readers need to see how the projected model is subjected to several security vulnerability tests. The mathematical proofs and validation criteria need to be well articulated.
Author Response
Reviewer#2, Concern #1:
- The authors have not addressed an earlier comment satisfactorily.
The performance analysis and security examinations lack the required rigors. We need to see how the projected model is subjected to several security vulnerability tests. The mathematical proofs and validation criteria need to be well articulated.
Authors response: We would like to clarify more about this concern and add more details as the reviewer comment have 2 parts.
Part 1: The performance analysis and security examinations lack the required rigors. We need to see how the projected model is subjected to several security vulnerability tests
We would like to highlight that the different types of attacks that the model is subject to, are already mentioned in section 4.2 “IoT-23 Dataset” as the traffic injected is typically a collection of different kind of attacks
DDoS: Denial of service attacks in which usually comes in volumetric order.
C&C traffic: command and control traffic either communicating periodically with a server or downloading malicious content, so it works in both directions of the traffic.
Okiru : which represents the Botnet traffic pattern, commonly used in IoT attacks.
Part of a horizontal scan: which is a common attack happening in the initial phases of reconnaissance.
The model was able to successfully classify the traffic either benign or Malicious (in the binary classification) and was able to classify the different types of attacks (in the Multiclass classification) with very high accuracy of 99.2% with the usage of filtered classifier algorithm. Both in binary and Multiclass classification results as in Table 5 and Table 8.
Authors action: We have updated the manuscript by adding a table summarizing the different types of attacks in the results section 5.2.2 Multiclass classification results - Table 10.
Part 2: The mathematical proofs and validation criteria need to be well articulated.
Authors response: The used validation criteria are standard formulas that are widely used in the literature. This part was addressed in the author action below by adding more clarifications.
Authors action: Regarding the mathematical proofs and Validation criteria, we’ve added in section “5.1 Evaluation metrics” the following clarification statement
The arguments used in these metrics were first presented in [25], other algorithms proposed in the literature [14] and [15], used the same evaluation metrics.
Reviewer 3 Report
The paper is acceptable in current form now as most of the suggestions have been incorporated in updated version.
Author Response
Reviewer#3 decision,
- The paper is acceptable in current form now as most of the suggestions have been incorporated in updated version
Authors response: We would like to thank the reviewer for his time taken to review the paper.
Round 3
Reviewer 1 Report
Unfortunately, I do not agree with the authors responses. IOT-23 is kind of an artificial dataset, a network with a few IOT devices, and some injected malicious agents which we hope the IDS can detect. Useful for testing what IDSes miss (false negatives), but hardly representative of EVCSes or any other network, for that matter. The fact that a supervised trained algorithm can achieve 99% performance when trained on that dataset doesn't really tell us anything. That a strategy of searching under lamp posts is effective at finding keys lost under lamp posts is not much use for finding keys that are more likely lost in the dark, away from a lamp post.
Anomaly detection may have a lot of "false positives", but those false positives are often very interesting events in themselves. For example, the event might be a router that has gone down, or congestion caused by a "Slashdot effect.
My recommendations remain the same.